# A Novel Magnetorheological Fluid with High-Temperature Resistance

**DOI:** 10.3390/ma16124207

**Published:** 2023-06-06

**Authors:** Jinjie Ji, Xiangfan Wu, Zuzhi Tian, Fangwei Xie, Fei Chen, Haopeng Li

**Affiliations:** 1School of Mechatronic Engineering, China University of Mining and Technology, Xuzhou 221116, China; 2School of Mechanical and Electrical Engineering, Xuzhou University of Technology, Xuzhou 221018, China

**Keywords:** magnetorheological fluid, high temperature, micro-nano, composite system

## Abstract

A magnetorheological fluid (MR fluid) is mainly composed of soft magnetic particles, surfactants, and the base carrier fluid. Among these, soft magnetic particles and the base carrier fluid influence the MR fluid significantly in a high-temperature environment. Therefore, a study was carried out to investigate the changes in the properties of soft magnetic particles and base carrier fluids in high-temperature environments. On this basis, a novel magnetorheological fluid with high-temperature resistance was prepared, and the novel magnetorheological fluid had excellent sedimentation stability, of which the sedimentation rate was only 4.42% after heat treatment at 150 °C followed by one-week placement. At 30 °C, the shear yield stress of the novel fluid was 9.47 kPa under the magnetic field of 817 mT: higher than the general magnetorheological fluid with the same mass fraction. Moreover, its shear yield stress was less affected by the high-temperature environment, reducing by only 4.03% from 10 °C to 70 °C. The novel MR fluid can be applied to a high-temperature environment, effectively expanding the application range of MR fluid.

## 1. Introduction

Magnetorheological (MR) materials are a significant branch of intelligent materials which make continuous and nonpolar reversible changes under the action of a magnetic field. MR fluids mainly comprise soft magnetic particles, surfactants, and the base carrier fluid [1,2,3,4,5]. Among these, the magnetic properties of soft magnetic particles are the main reason for the unique rheological effect of MR fluids. Furthermore, the high-temperature oxidation properties of soft magnetic particles also directly influence transmission factors such as flow characteristics, sedimentation stability, and the shear yield stress of MR fluids. The base carrier fluid type has an important influence on performance indexes such as the working temperature and zero-field viscosity of MR fluids [2].

Currently, MR fluid, the most widely studied and applied MR material, is commonly used in active vibration control [3,4,5,6,7], transmission [8,9], braking [10,11], polishing [12,13], sealing [14,15], suspensions [16], haptic modules [17,18], and so on [19,20,21,22,23]. However, problems with temperature rises are inevitable in most MR devices. The high-temperature environment caused by the frictional heat between particles in the MR fluids and the electromagnetic heat of the coil will accelerate the oxidation and sedimentation of the soft magnetic particles in the MR fluids. This situation will aggravate the sedimentation of MR fluids, weaken shear yield stress, and reduce the service life, which will eventually reduce the stability and reliability of MR devices.

In order to solve the above issues, many scholars have conducted a great deal of research on the structure of magnetorheological fluid transmission devices to enhance the heat dissipation capability of the devices themselves. Wang et al. [24] designed a liquid-cooled high-power clutch and proved that the liquid cooling method could effectively improve the heat dissipation of the clutch. On this basis, Wu et al. [8] designed a hollow-structure water-cooled MR clutch to further enhance the heat dissipation effect of the device. Huang et al. [25] designed a multi-channel cooling structure to enhance the cooling efficiency of MR devices. Arief [26] found that a high temperature will cause thinning of the MR fluid. The MR fluid prepared by Tian et al. [27] using fluorine oil and a coupling agent as surfactant still performs well at high temperature. Thakur, M. K. [28] studied graphene’s effect on MR fluid in terms of high-temperature stability and shear yield stress.

However, fewer scholars are conducting research based on the high-temperature characteristics of the components of MR fluids to improve the overall high-temperature stability of the fluids. In this work, the magnetic properties and high-temperature oxidation of soft magnetic particles in a high-temperature environment were analyzed, and the viscosity, thermal evaporation, and thermal expansion properties of the base carrier fluids under a high-temperature environment were analyzed. The MR fluid with excellent high-temperature stability was studied by selecting high-performance materials, and its various properties were tested. This work provides an excellent MR fluid for MR devices in high-temperature conditions, which can enhance the application prospects of MR fluid.

## 2. Materials and Methods

### 2.1. Soft Magnetic Particles

The magnetic properties of soft magnetic particles are the main reason for the unique rheological effect of MR fluids. In a common preparation process, micron-sized carbonyl iron powder (CIP) is often used as a source of soft magnetic particles to prepare MR fluid, due to its simple preparation, low price, high purity, and fine particle size. However, the high-temperature stability and shear yield stress of magnetorheological fluids prepared by CIP make it challenging to meet the usage requirements. It has been found that doping a small amount of soft magnetic nanoparticles in micron-sized CIP particles can effectively improve the sedimentation stability and shear yield stress of MR fluid [29]. Furthermore, the selected nanoparticles should have good magnetic saturation strength and high Curie temperature, considering the demand for high-temperature transmission conditions. Fe-Co-Ni nanoparticles possess higher magnetic saturation intensity (2.4 kA/m), and their Curie temperature of 980 °C is higher than that of CIP. In this work, a small number of Fe-Co-Ni nanoparticles were selected to form a micro-nano composite system by compounding with CIP particles. The microparticles in the micro-nano composite system were CIPs of 2 μm supplied by TIANYI (China), and Fe-Co-Ni nanoparticles of 100 nm were supplied by QIYUE (China).

The MPMS 3 by QUANTUM DESIGN (America) was applied to measure the magnetic properties of the micro-nano composite system and CIPs in a high-temperature environment. As shown in Figure 1, the magnetization intensity of particle groups decreases with increasing temperature. The magnetic saturation intensity of the micro-nano composite system is higher than that of the CIPs in both 46.85 °C and 246.85 °C cases. The saturation magnetization intensity of the micro-nano composite system is 193.93 emu/g at 246.85 °C, which can provide greater shear yield stress. When the temperature increases from 46.85 °C to 246.85 °C, the magnetic saturation intensity of the micro-nano composite system changes the least, with only a 5.5% decrease.

Moreover, it can be seen from Figure 1 that the coercivity of soft magnetic particles decreases significantly with the increase in temperature. During the temperature increase from 46.5 °C to 246.5 °C, the coercivity of Sample A is reduced from 4.6 kA/m to 0.58 kA/m, with a more significant decrease of 87.4%. The coercivity of Sample B is reduced from 0.887 kA/m to 0.3193 kA/m, representing a 64% reduction. The composite systems composed of CIPs and nanoparticles exhibit low coercivity and relatively small dropouts at room and high temperatures. This indicates a good enhancement effect on the preparation of magnetorheological fluids. 

Figure 2 shows that the soft magnetic particles should also have a good high-temperature stability. The Q5000IR by TA (America) was applied to measure the thermalgravimetric analysis of the micro-nano composite system. When the temperature is in the range of 25~300 °C, the mass percentage of the micro-nano composite system shows a downward trend. In addition, the maximum mass percentage is 98.7% at 300 °C. This is due to the nanoparticles containing a small amount of deionized water that evaporates as temperatures rise. In the temperature range of 25~300 °C, there is no oxidation or weight gain in the samples. Generally speaking, the working temperature of MR fluid under transmission conditions is far below 300 °C, meeting the functional requirements.

### 2.2. Base Carrier Fluid

The expansion, evaporation, and viscous temperature characteristics of the base carrier fluid significantly impact the performance of MR fluid in a high-temperature environment. In this work, the six different types of high-temperature resistant base carrier fluids in Table 1 were tested.

#### 2.2.1. Temperature–Viscosity Characteristics

The SNB-1T digital rotational viscometer by FANGRUI (China) and the digital display thermostatic oil bath were applied to measure the viscosity–temperature characteristics of the base carrier fluids in Table 1.

As shown in Figure 3, the dynamic viscosity of the six samples has a linear relationship with temperature. The dynamic viscosity of PMX-200, PMX-0156, and Grape seed oil decreases sharply in the range of 10–60 °C and the decrease slows as the temperature approaches 70–180 °C. The dynamic viscosity of RSF-102, FL310, and Castor oil (at room temperature) decreases more sharply than the others. The silicone oils have better performance than the other kinds of base carrier fluids. When the temperature is raised from 10 °C to 140 °C, the viscosity of the samples decreases by 82%, 0.91%, 0.998%, 0.994%, 0.944%, and 0.994%.

When the temperature rises, the arrangement of molecular spacing within the fluid is damaged, the kinetic energy between fluid molecules increases, the molecular spacing becomes more significant, the internal friction resistance decreases, and the viscosity of the liquid decreases. The viscosity index is commonly used to describe the degree to which the viscosity of a fluid changes with temperature. The higher the viscosity index, the less the viscosity of the fluid is affected by temperature. The viscosity indices of the six samples can be obtained from Figure 4, and the viscosity indices of the six base carrier fluids are PMX-200 (430) > PMX-0156 (359) > Grape seed oil (125) > FL310 (85) > RSF-102 (44) > Castor oil (37). It can be seen that the viscosity indices of PMX-200 and PMX-0156 are 344% and 287.2% of that of Grape seed oil, respectively, with excellent high-temperature stability.

RSF-102, FL310, and Castor oil have dynamic viscosities of 4.07 Pa·s, 2.08 Pa·s, and 1.248 Pa·s at 20 °C, respectively. The dynamic viscosities of these base carrier fluids are relatively high, which should be discussed further with subsequent experiments.

#### 2.2.2. Expansion Characteristics

The thermal expansion property of base carrier fluids refers to the phenomenon that the volume of the fluids expands with the increasing external temperature under constant external pressure. The volume expansion coefficient is often used to characterize the expansion properties of liquid substances, and can be represented as
(1)$$\alpha_{v}=\frac{V_{2}-V_{1}}{V_{1}(T_{2}-T_{1})}$$
where *V*_1_ and *V*_2_ are the volume of the liquid at temperatures *T*_1_ and *T*_2_, respectively, as can be seen in Figure 5.

It can be seen from Figure 6 that when the temperature ranges from 10 °C to 250 °C, the expansion volumes of the base carrier fluids increase with the increasing temperature, and the relationship is approximately linear. In the range of 10 to 210 °C, the expansion rates of the six samples are 18.4%, 20%, 16%, 18%, 15%, and 12%, respectively. Although the expansion rates of PMX200 and PMX-0156 are larger, silicone oil has a stable change with the thermal expansion coefficient, which remains at about 0.95 × 10^−3^/°C in the range of 10 to 200 °C.

#### 2.2.3. Evaporation Characteristics

The expansion rate of base carrier fluids can be obtained from Equation (2),
(2)$$r_{ev}=\frac{W_{\mathrm{ev}}}{W_{o}}\times 100\%$$
where *W_ev_* is the recorded weight after heating and *W_o_* is the original weight before heating.

It can be seen from Figure 7 that Grape seed oil has the most stable volatilization rate among all base carrier fluids. The specific evaporation rates of each base carrier fluid are as follows: Grape seed oil (0%), PMX-200 (100 cSt) (0.61%), FL310 (1.78%), PMX-0156 (2.1%), and Castor oil (2.5%). The maximum evaporation rate is observed in RSF-102 (3.3%).

The content of the above research is summarized in Table 2. The performance of samples was ranked from 1 to 6, with 1 indicating the best performance and 6 indicating the worst performance. PMX-200, PMX-0156, and Grape seed oil had the best all-around performance among the six base carrier fluids. However, the main component of Grape seed oil is linoleic acid, which is prone to carcinogenic smoke and harmful to human health at high temperatures. In the following research, the preparation of MR fluid will be further studied by using PMX-200 and PMX-0156 as the base carrier fluids.

### 2.3. Preparation Methods

The preparation process of MR fluids is shown in Figure 8. The soft magnetic particles and surfactants were thoroughly mixed in anhydrous ethanol by a stirrer, ultrasonic disperser, and ball mill. The surfactant completely enveloped the soft magnetic particles. The whole wrapping process lasted 4 h and was carried out at 40 °C. Then, the mixture was dried in a vacuum drying oven at 80 °C for three hours. After that, the mixture was dispersed using a ball mill, and the modified soft magnetic particles were obtained. Finally, the modified particles, hydrophobic fumed silica and dimethyl silicone oil were thoroughly mixed by high-speed stirring and ultrasonic dispersion for 4 h. The soft magnetic particles were uniformly dispersed in the base carrier fluid. The whole dispersion process was carried out using an oil bath to ensure a constant temperature of 40 °C.

## 3. Results

### 3.1. Sedimentation Rate

According to the previous study, two formulations of MR fluids were prepared and placed under different working conditions to measure their sedimentation rates. The specific formulations of MR fluids are shown in Table 3. The content of nanoparticles in the micro-nano composite system is 10%. Formulation A is composed of 1.5 wt% sodium lauryl phosphate, 1.0 wt% complex MES, 1.5 wt% ethylene glycol monostearate, 1.5 wt% glycerol monostearate, and 1.0 wt% hydrophobic pyrogenic silica. In addition, the high-temperature working conditions are shown in Table 4.

As shown in Figure 9, the sedimentation stability of Sample A deteriorated sharply with increasing temperature. The seven-day maximum sedimentation rate of Sample A at room temperature was 0.99%. The seven-day maximum sedimentation rate was 1.86% in working condition 3, which was about 188% of that at room temperature. The maximum seven-day sedimentation rate of Sample A was 4.42% in working condition 5.

The variation of the sedimentation stability for Sample B is quite interesting. At room temperature, the sedimentation rate of Sample B remained constant after three days. Multiple experimental verifications revealed that the upper part of Sample B formed an odorless solidified black colloid with time, leading to the inability to observe its sedimentation rate. Additionally, the high-temperature conditions accelerated this chemical reaction, resulting in the sedimentation rate of Sample B in working condition 4 and working condition 6 being 0%.

Hydroxy silicone oil reacts chemically with some surfactants at room temperature, causing the liquid to form a colloid. High-temperature conditions can provide a better catalytic effect for this chemical reaction. In this study our aim was to prepare a high-temperature-resistant MR fluid for transmission equipment. Hence, formulation B is unsuitable.

### 3.2. Expansion Rate

Sample A and dimethyl silicone oil were measured in a high-temperature environment of about 250 °C for 4 h.

As shown in Figure 10, the evaporation rate of dimethyl silicone oil was 0.6134%, and that of Sample A was 0.354% in the same experimental conditions. Among the components of the MR fluid, the soft magnetic particles are denser and have a higher Curie point, making them less prone to chemical reactions at 250 °C. The overall content of surfactants was low and negligible. The base carrier fluid is prone to evaporation in a high-temperature environment, and the weight loss of the MR fluid is mainly attributable to the evaporation of the base carrier fluid. At 250 °C, the micro-nano composite system was almost entirely free from oxidation, and the prepared samples had good high-temperature stability.

### 3.3. Volatilization Rate

Zero-field viscosity is an essential indicator in MR fluid transmission. However, it decays considerably at high temperatures. Therefore, a comparative test was conducted to analyze the effect of CIPs and Fe-Co-Ni nanoparticles on the viscosity–temperature characteristics of MR fluids without changing other parameters in the formulation and only changing the components of soft magnetic particles. The effect of CIPs and Fe-Co-Ni nanoparticles on the viscosity–temperature properties of MR fluids was analyzed by a comparative test.

It can be seen from Figure 11 that the expansion rate of MR fluids gradually increased with increasing temperature, and the expansion trend was similar to that of dimethyl silicone oil. When the temperature was 250 °C, the MR fluid expansion rate reached 10.53%. This was mainly due to the influence of temperature on the base carrier fluid, whose atomic structure spacing increased with enhanced thermal motion and liquid volume expansion. After the temperature reached 210 °C, the expansion rate of both samples stopped increasing, mainly due to the thermal evaporation of the base carrier fluid.

### 3.4. Viscosity–Temperature Properties

A comparison group was also set up to analyze the trend of high-temperature performance of the novel MR fluid. The comparison group formulations of Sample C are composed of sodium lauryl phosphate (1.5 wt%), complex MES (1.5 wt%), ethylene glycol monostearate (1.5 wt%), glycerol monostearate (1.5 wt%), hydrophobic pyrogenic silica (1.5 wt%), CIPs (40 wt%), and dimethyl silicone oil (53.5 wt%).

The SNB-1T digital rotational viscometer by FANGRUI (China) was applied to measure the temperature–viscosity properties of Sample A and Sample C.

The variation pattern of the viscosity–temperature curve for MR fluids is similar to that of the base carrier fluid. At the same time, it can be seen from Figure 12 that the zero-field viscosity of Sample C is lower than that of Sample A, and its temperature change curve is smoother. The viscosity of MR fluids changes sharply when the temperature drops from 100 °C to 20 °C; that of Sample A changes from 0.522 Pa·s at 100 °C to 1.782 Pa·s at 20 °C, and that of Sample C from 0.633 Pa·s at 100 °C to 1.984 Pa·s at 20 °C. This is mainly due to the smaller average particle size of the micro-nano composite system in Sample C compared to that of the CIPs, and the smaller particle size helps to improve the zero-field viscosity of the magnetorheological fluid. The high temperature decreases the zero-field viscosity of MR fluids. The increasing temperature will lead to increased thermal movement of molecules inside the liquid, which decreases the zero-field viscosity. The zero-field viscosity of Sample A is higher than that of Sample C in the same working conditions. The zero-field viscosity of both samples continues to decrease as the temperature increases. The high temperature will reduce the viscosity of MR fluid, which is undoubtedly a favorable change.

### 3.5. Shear Yield Stress

The rotational rheometer MCR 301 equipped with MRD 180 by ANTON PAAR (Austria) was used to obtain the rheological curves of Sample A and the general MR fluid (Sample C). The Herschel–Bulkley model with better fitting effects [30] was applied to obtain the shear yield stress of the MR fluid, as shown in Figure 13.

At 30 °C, the yield stress of Sample A was sharply enhanced before 500 mT, but the increase decreased after reaching a specific value. Sample A and Sample C both reached the limit at 817 mT with 9.47 kPa and 7.23 kPa, respectively. The shear stress of Sample A was about 131% that of Sample C. This experiment proves that the micro-nano composite system could effectively enhance the shear yield stress of the MR fluid.

The effect of temperature on shear yield stress is shown in Figure 14. Due to the performance of the equipment, the maximum temperature can only be detected at 70 °C. The shear stress of Sample A is better than that of ordinary MR fluid at different temperatures. The shear yield stress performance of Sample A increases linearly between 20 and 40 °C. The maximum shear stress of the MR fluid appears at 40 °C. Subsequently, the MR fluid yield stress decays further with increasing temperature.

From the above analysis, it is clear that the properties of soft magnetic particles mainly influence the yield stress of MR fluids. The MR fluid comprising the micro-nano composite system has better yield stress.

### 3.6. Response Time

In order to observe the microscopic manifestation of the rheological effect of magnetorheological fluid and the time response characteristics, a high-speed photomicrographic imaging test bench was established, as shown in Figure 15. The high-speed photomicrographic imaging test bench was mainly composed of a high-speed camera, microlens, and magnetic field generator. The specific response time of the magnetorheological fluid particles under the action of a magnetic field could be obtained by the joint triggering of a high-speed camera and a magnetic field generation device. In Figure 16, it can be seen that the particles were rapidly arranged in chains along the direction of the magnetic field, and the response speed was fast. The particles reached the chain state at about 10 ms, and the response time changed on the order of milliseconds. The response time of the prepared MR fluids was excellent.

## 4. Conclusions

In this work, the properties of soft magnetic particles and base carrier fluids for MR fluids were investigated in a high-temperature environment. The micro-nano composite system and dimethyl silicone oil show an excellent performance at high temperatures. The micro-nano composite system was composed of CIPs supplied by TIANYI (China) and Fe-Co-Ni supplied by QIYUE (China), and its mass fraction was 40 wt%. The surfactants and thixotropic agents were sodium lauryl phosphate (1.5 wt%), complex MES (1.0 wt%), ethylene glycol monostearate (1.5 wt%), glyceryl monostearate (1 wt%), and hydrophobic fumed silica (1.0 wt%). The base carrier fluid was 100 cSt dimethyl silicone oil. According to above formulation, a novel MR fluid for high temperatures was prepared and its sedimentation stability, expansion rate, volatilization rate, and temperature–viscosity property, as well as shear yield stress were investigated at high temperatures. In the experiments, the novel MR fluid had better sedimentation stability and higher shear yield stress.

## Figures and Tables

**Figure 1 materials-16-04207-f001:**
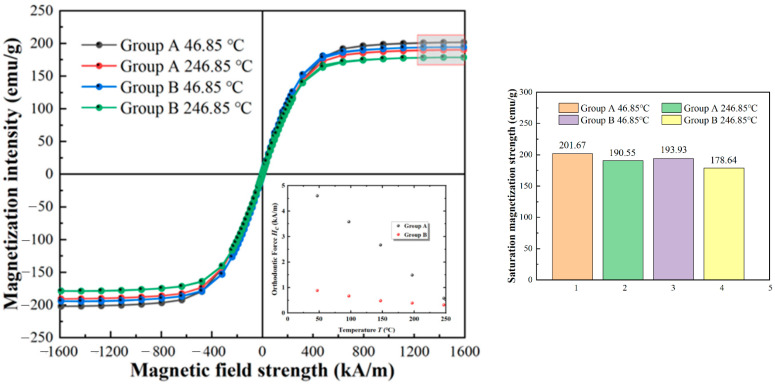
The hysteresis curve of soft magnetic particles. Group A: micro-nano composite system; Group B: CIPs.

**Figure 2 materials-16-04207-f002:**
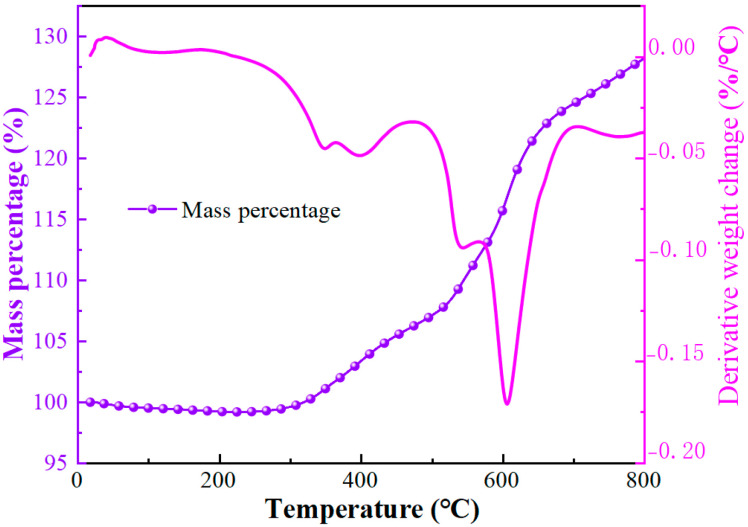
The TGA curve of soft magnetic particles from 25 to 800 °C.

**Figure 3 materials-16-04207-f003:**
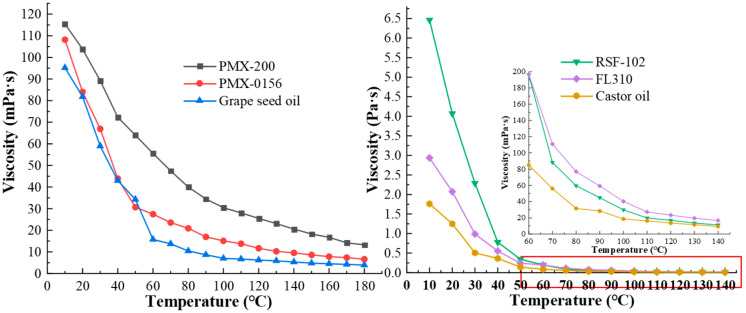
The viscosity–temperature curves of six base carrier fluids.

**Figure 4 materials-16-04207-f004:**
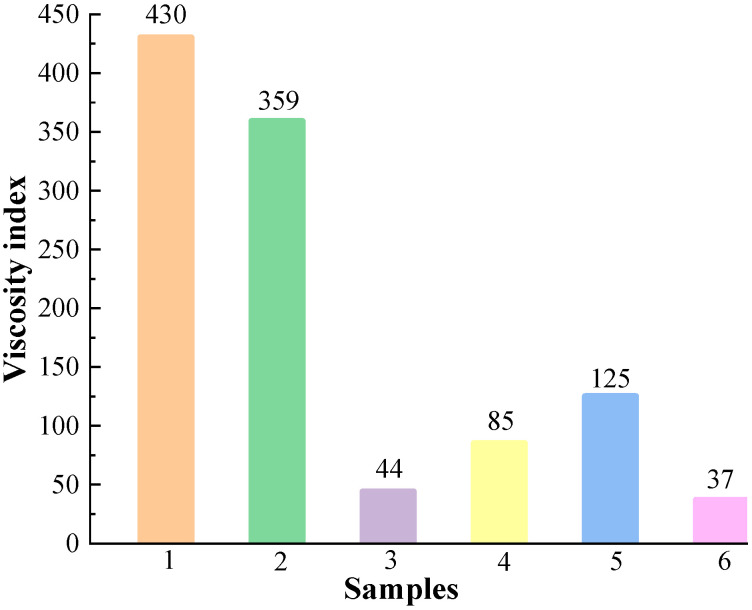
The viscosity indices of base carrier fluids.

**Figure 5 materials-16-04207-f005:**
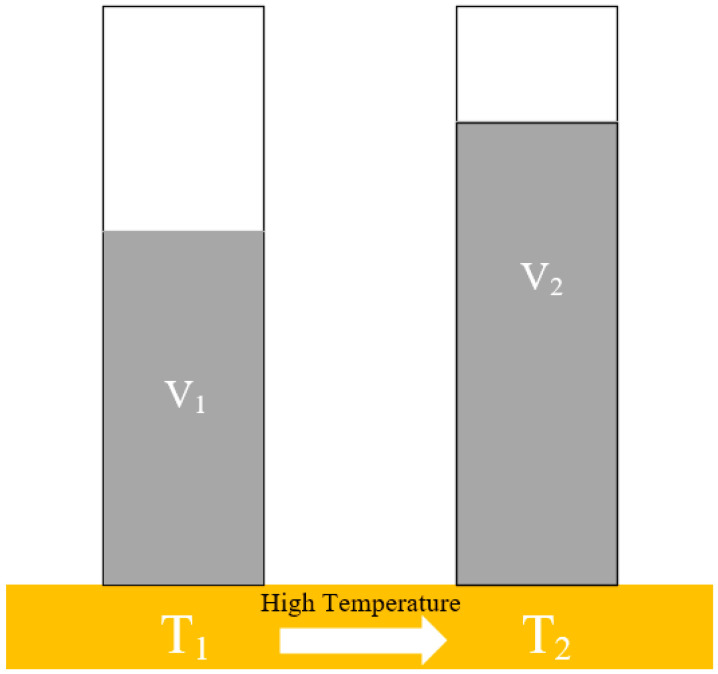
The expansion rate diagram of base carrier fluids.

**Figure 6 materials-16-04207-f006:**
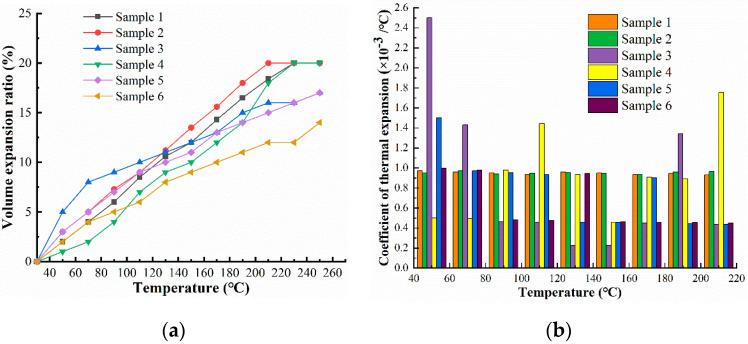
The volume expansion rates of base carrier fluids. (**a**) The curve of temperature and volume expansion ratio (**b**) Coefficient of thermal expansion.

**Figure 7 materials-16-04207-f007:**
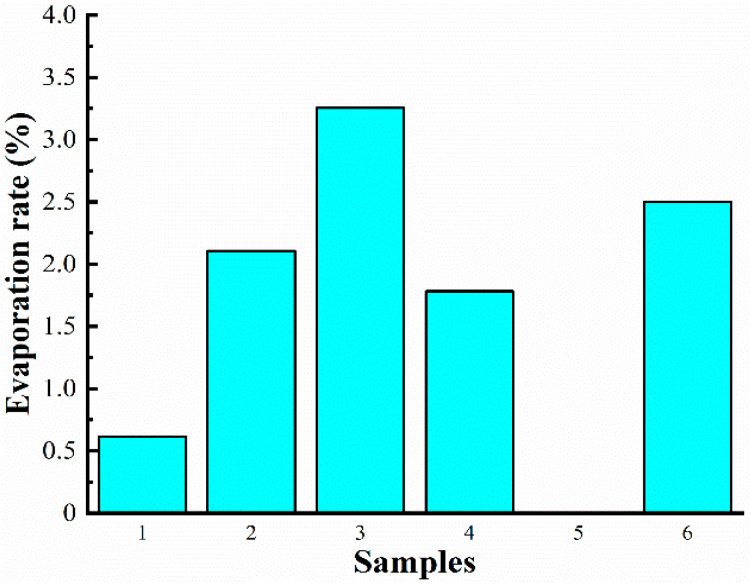
The evaporation rates of samples in a high-temperature environment.

**Figure 8 materials-16-04207-f008:**
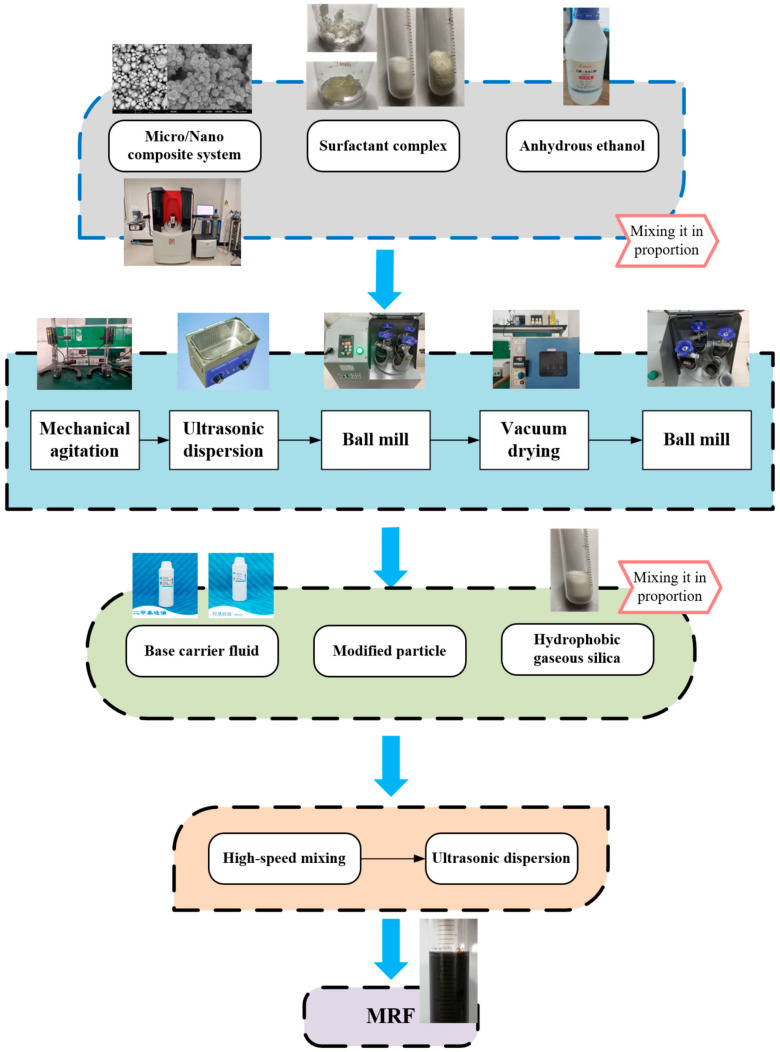
The preparation process of the novel MR fluid.

**Figure 9 materials-16-04207-f009:**
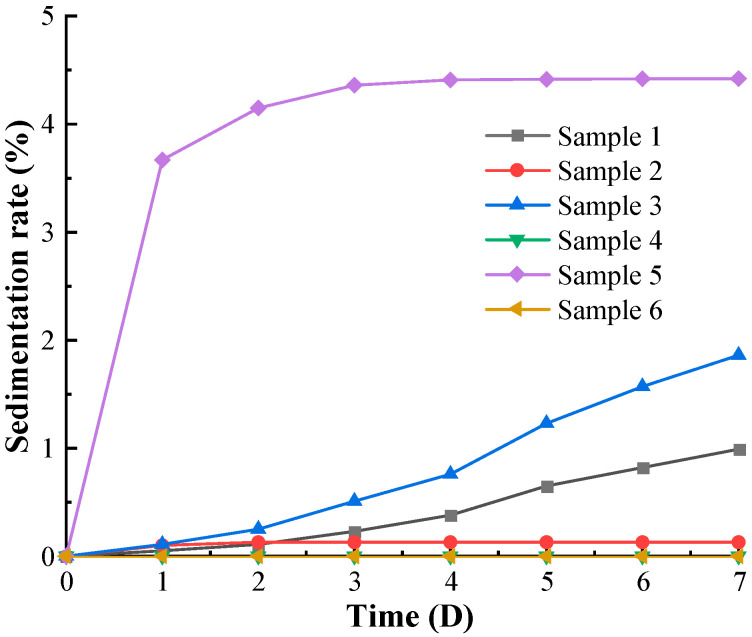
The sedimentation curves of MR fluids with different working conditions.

**Figure 10 materials-16-04207-f010:**
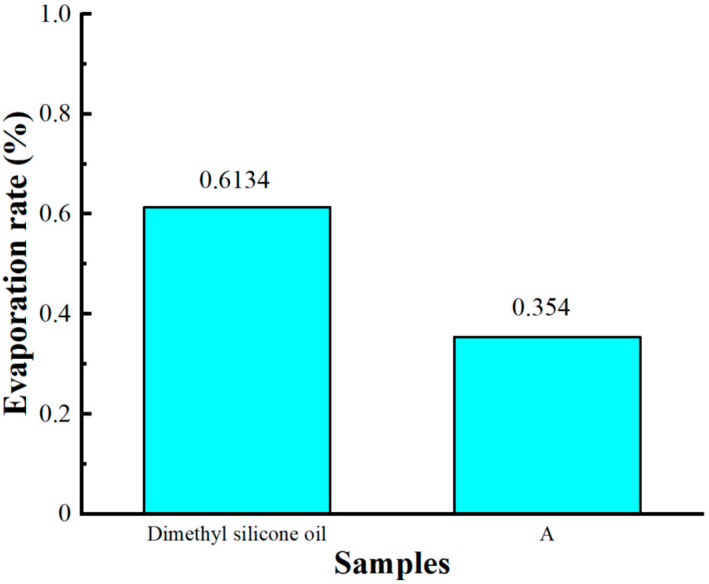
The evaporation rate of Sample A and dimethyl silicone oil.

**Figure 11 materials-16-04207-f011:**
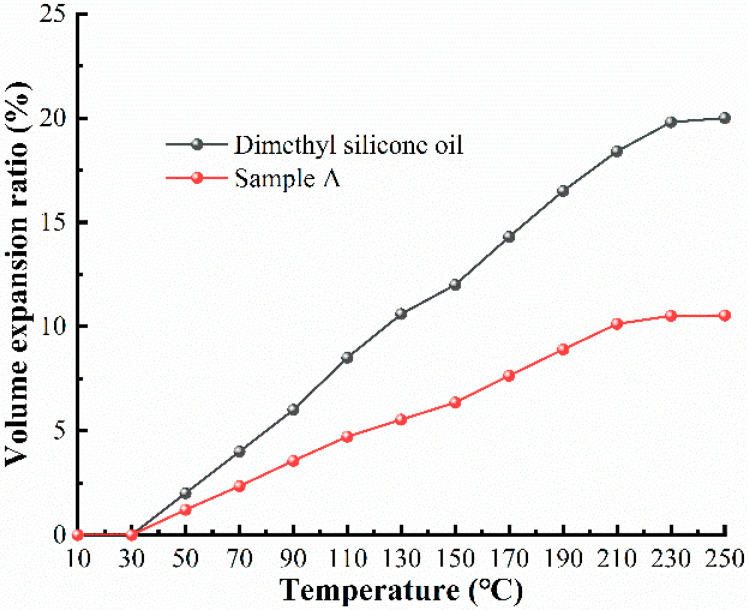
The expansion rate of Sample A at different temperatures.

**Figure 12 materials-16-04207-f012:**
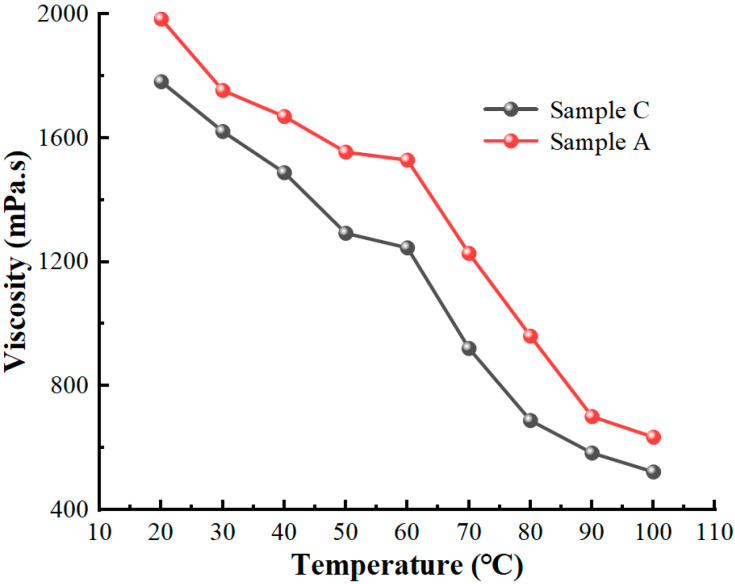
The temperature–viscosity properties of MR fluids at different temperatures.

**Figure 13 materials-16-04207-f013:**
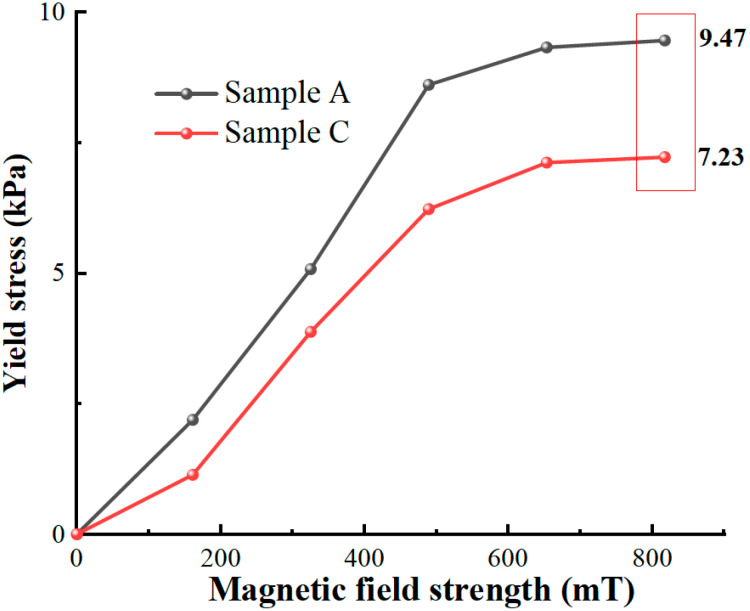
The yield stress of MR fluids at 30 °C with different magnetic field strengths.

**Figure 14 materials-16-04207-f014:**
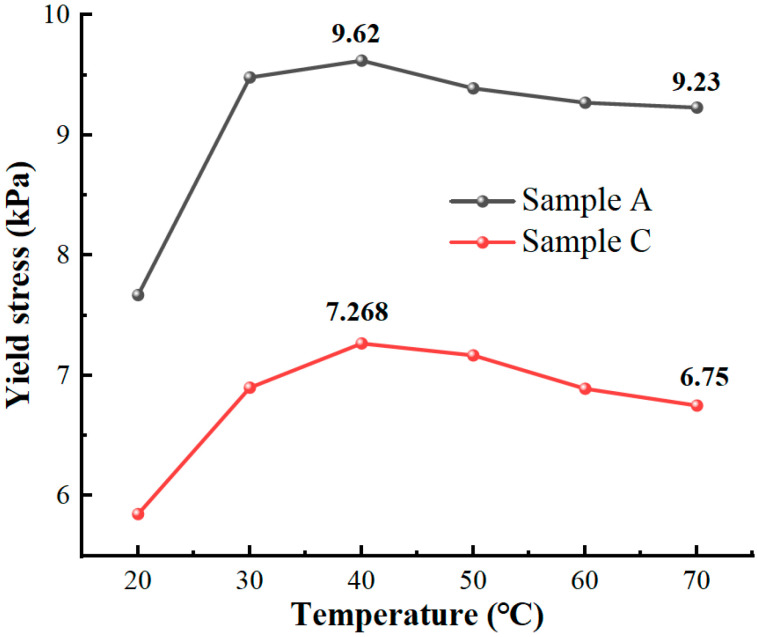
The magnetic saturation yield stress of MR fluids at different temperatures.

**Figure 15 materials-16-04207-f015:**
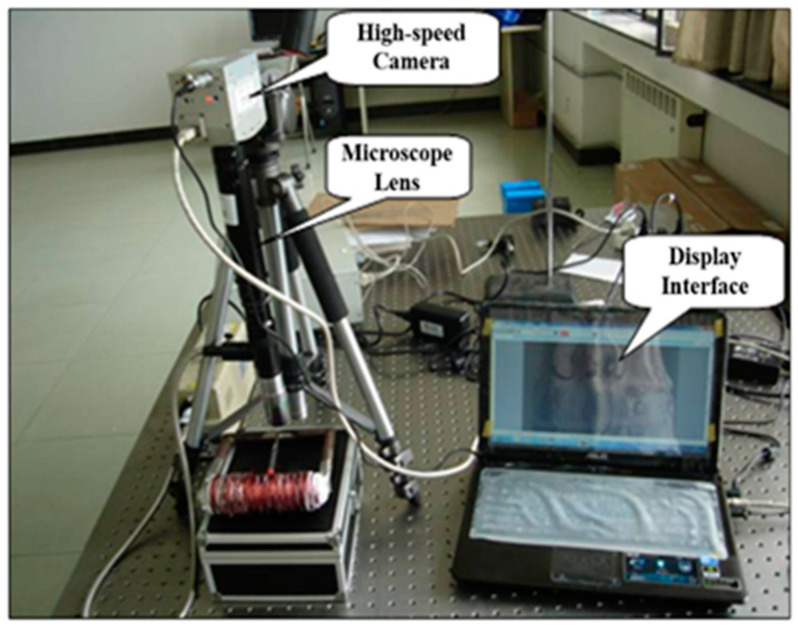
The high-speed photomicrographic imaging test bench.

**Figure 16 materials-16-04207-f016:**
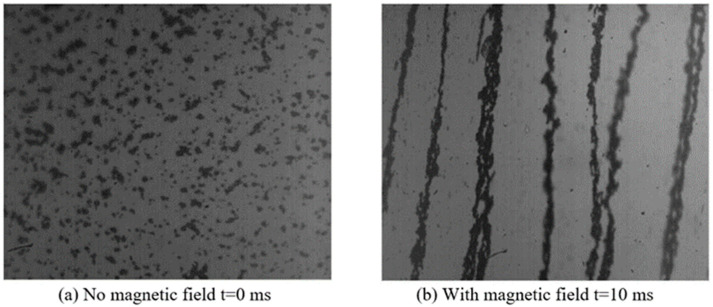
The response time of the prepared MR fluids under different magnetic fields.

**Table 1 materials-16-04207-t001:** The information on base carrier fluids.

Number	Name	Company (Country)	Main Components	Max Temperature (°C)
1	PMX-200 (100 cSt)	DOW CORNING (America)	Polydimethylsiloxane	300
2	PMX-0156	ECOSOL (China)	Siloxane and polysiloxane	318
3	RSF-102	RUNXING (China)	Thickened by thickener	350
4	FL310	AIDI (China)	Polytetrafluoroethylene	300
5	Grape seed oil	ECOSOL (China)	Linoleic acid	280
6	Castor oil	ADANI (India)	Triglycerides	313

**Table 2 materials-16-04207-t002:** Performance parameters comparison of the base carrier fluids.

Name	Viscosity–Temperature Characteristics	Expansion Rate	High-Temperature Performance	Evaporation Rate
PMX-200	**1**	4	**2**	**2**
PMX-0156	**2**	5	**1**	4
RSF-102	5	**3**	5	6
FL310	4	6	4	**3**
Grape seed oil	**3**	**2**	**3**	**1**
Castor oil	6	**1**	6	5

**Table 3 materials-16-04207-t003:** The specific formulation of MR fluids.

Number	Soft Magnetic Particles (40 wt%)	Surfactants (6.5 wt%)	Base Carrier Fluid (53.5 wt%)
A	CIPs and Fe-Co-Ni nanoparticles	Formulation A	Dimethyl silicone oil
B	CIPs and Fe-Co-Ni nanoparticles	Formulation A	Hydroxy silicone oil
C	CIPs	Formulation A	Dimethyl silicone oil

**Table 4 materials-16-04207-t004:** The experimental methods for MR fluid samples.

Number	Experimental Methods
1	Sample A at normal temperature condition
2	Sample B at normal temperature condition
3	Sample A heat for 1 h at 60 °C
4	Sample B heat for 1 h at 60 °C
5	Sample A heat for 1 h at 150 °C
6	Sample B heat for 1 h at 150 °C

## Data Availability

Data sharing is not applicable for this article.

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
