# Peer review of "A Novel Magnetorheological Fluid with High-Temperature Resistance"

_materials, 2023, doi:10.3390/ma16124207_

Round 1

Reviewer 1 Report

The authors prepared a novel magnetorheological fluid with high-12 temperature resistance and investigated the changes in the properties of soft magnetic particles and base carrier fluids under high-temperature environments.  This manuscript is nice and well written, however, there are some points to be discussed.

1.     Introduction,

 Nowadays, as the authors mentioned in Introduction, MR fluids have been widely used for many devices and applications such as active vibration control, transmission, braking, polishing, and sealing. Besides these applications, Recently, MR fluids are applied for suspensions and haptic modules.

The authors had better insert these two applications in introduction and refer some previous works (following research works).

1)    A review of the state of the art in magnetorheological fluid technologies--Part I: MR fluid and MR fluid models, FD Goncalves, JH Koo, M Ahmadian - The Shock and Vibration Digest, 2006   

2)    A tiny haptic knob based on magnetorheological fluids, YH Heo, DS Choi, IH Yun, SY Kim - Applied Sciences, 2020 - mdpi.com

3)    Smart Design of Z-Width Expanded Thumb Haptic Interface Using Magnetorheological Fluids, X Yin, C Wu, S Wen, J Zhang - IEEE Transactions on , 2021 - ieeexplore.ieee.org

2.     How quickly does the prepared MR fluids become solidify? The authors should investigate response time of the prepared MR fluids using an experiment.

3.     There can be some precipitation in MR fluids if the MR fluids does not work. The authors had better show the amount of precipitation of MR fluids. The reviewer knows the fact that there is precipitation. However, the reviewer wants to know the amount of the precipitation.

Author Response

Dear editor,

Thank you for your efforts on our manuscript.

Based on the reviewer’s comments, we had a major revision on the previous manuscript, and the latest version was attached. Besides, we give a response to the reviewer’s comments item by item, which are as follows:

Reviewer 2 Report

The authors performed an extensive study on high temperature characteristics of MR fluids to engineer a state of the art MR fluid composition. The authors investigated dynamic material properties of components of MR fluids in various temperature conditions which are structurally or compositionally relevant (corresponding to a structural or compositional change that might occur due to temperature conditions). This is a rather unique approach considering development of MR fluids, but quite common in composite material design. This paper is prepared and penned scientifically sound. However, there are several parts that can help improve the quality and clarity of this work:

1- Figure affiliations are rather missing or not labeled (i.e. as shown in figure ?, question mark indicates it is not labeled in accordance with relevant figure). This lack of clarity disperses the focus of the science.

2- The authors performed magnetic property discussion simply based on saturation magnetization. Saturation magnetization definitely can indicate a structural and compositional change, however a direct measure of loss (magnetic or resistive) in magnetic systems can be acquired by investigating coercivity of the magnetic material systems. So, I would very much like to see a detailed analysis on coercivity, which can be acquired by Figure 1a. An expanded version of Figure1a around 0 magnetization intensity as an inset can help visualization of this analysis.

3- There is two distinct sets of Sample 1 to 6, please use a different wording to clarify this confusion. Also, in the middle of the text you introduce a Sample C to the discussion, please add sample C to Table 3. Also explain the formulation of Sample C earlier and provide a merit for using this composition as somewhat a control sample to evaluate the performance of your MR fluids.

4- This work can definitely benefit from a concentration essay (various concentrations of solid content in engineered MR fluids) to evaluate the uppermost potential of this study. But, I will understand it might be beyond the scope of this manuscript.

Besides these points, I believe this a sound work that offers a different perspective (focused on materials science) on MR fluid systems.

Author Response

(The authors gave the same response as above.)
